# SBI-RAG: Enhancing Math Word Problem Solving for Students through Schema-Based Instruction and Retrieval-Augmented Generation

**Prakhar Dixit**
Department of Computer Science
University of Maryland Baltimore County
`pdixit1@umbc.edu`

**Tim Oates**
Department of Computer Science
University of Maryland Baltimore County
`oates@cs.umbc.edu`

## Abstract

Many students struggle with math word problems (MWPs), often finding it difficult to identify key information and select the appropriate mathematical operations. Schema-based instruction (SBI) is an evidence-based strategy that helps students categorize problems based on their structure, improving problem-solving accuracy. Building on this, we propose a Schema-Based Instruction Retrieval-Augmented Generation (SBI-RAG) framework that incorporates a large language model (LLM). Our approach emphasizes step-by-step reasoning by leveraging schemas to guide solution generation. We evaluate its performance on the GSM8K dataset, comparing it with GPT-4 and GPT-3.5 Turbo, and introduce a "reasoning score" metric to assess solution quality. Our findings suggest that SBI-RAG enhances reasoning clarity and facilitates a more structured problem-solving process potentially providing educational benefits for students.

## 1   Introduction

Proficiency in solving math word problems (MWPs) is not only measured by students' ability to arrive at the correct solution but also by their capacity to follow a structured, step-by-step reasoning process [8]. This approach is vital for developing critical thinking and mathematical reasoning abilities, which are essential for tackling complex word problems effectively [18]. Unfortunately, many students struggle with word problems, often failing to identify key information or select the appropriate operations despite understanding the underlying mathematical concepts. This difficulty is a significant barrier to academic success, as highlighted by a survey from the EdWeek Research Center, which reported that nearly 50% of students can read a word problem's text but fail to grasp the mathematical question being asked [24]. Consequently, poor problem-solving skills in MWPs can lead to academic challenges and even failure in school.

Word problems, as a core component of the mathematics curriculum, serve an important function by fostering logical analysis, mental abilities, and creative thinking. Educators and researchers have explored various methods to improve students' proficiency in solving these problems. One such method is Schema-Based Instruction (SBI) [27, 7], an evidence-based approach widely used in the field of Mathematics that helps students classify word problems based on their underlying structure or schema. SBI has been shown to enhance students' ability to identify relevant information and apply the appropriate mathematical operations for problem-solving. In addition to educational approaches like SBI, Intelligent Tutoring Systems (ITSs) have emerged as valuable tools in addressing challenges associated with MWPs. ITSs leverage artificial intelligence (AI) and interactive interfaces to provide personalized, step-by-step guidance. Examples of ITSs designed for word problem-solving include AnimalWatch [2], MathCAL [4], PAT (Pump Algebra Tutor) [15] and HINTS [35]. These systems have proven effective in supporting learners by offering feedback, hints, and individualized learning paths. However, many ITSs rely on rule-based algorithms and lack the transformative potential of more recent AI advancements, like those in Natural Language Processing (NLP) [19] and the

38th Conference on Neural Information Processing Systems (NeurIPS 2024).

development of Large Language Models (LLMs), such as ChatGPT [3], LLaMA 2 [28] and Gemini [26].

LLMs exhibit emergent abilities, such as understanding linguistic nuances, making logical inferences, and decomposing tasks into simpler steps, which can be harnessed to scaffold students' learning in math-related tasks [12]. However, while LLMs like GPT-4 and others can generate intermediate steps through approaches like Chain-of-Thought (CoT) prompting [33], this happens primarily at the prompting level and requires the user to have knowledge of how to effectively structure the thoughts. CoT prompting is highly dependent on precise prompt engineering; poorly designed or unclear prompts can result in irrelevant or inefficient reasoning steps. Additionally, CoT prompting can sometimes generate illogical chains of thought, exposing gaps in the reasoning process [22]. Creating effective CoT prompts can be time-consuming and complex, particularly for advanced tasks.

In this paper, we propose a Schema-Based Instruction Retrieval-Augmented Generation (SBI-RAG) framework incorporating a Large Language Model (LLM) to assist in solving MWPs. Our system first utilizes a schema classifier, trained on DistilBERT [23], to predict the appropriate schema $S_i$ for a given problem $P$. The identified schema is then used to generate a schema-specific prompt, which retrieves relevant context from a pre-defined document set. The retrieved context, schema, and problem are passed to an LLM (Ollama Llama 3.1), which generates a detailed, step-by-step solution.

Our findings suggest that the schema-guided RAG approach facilitates a more structured problem-solving process, which we believe will lead to improved reasoning and deeper student understanding of MWPs. This framework also forms a pathway for future work, where we can incorporate feedback from teachers and students to refine and adapt the system further, thereby improving reasoning and critical thinking skills. By leveraging this system in classroom settings, we can iteratively enhance its effectiveness as both a teaching and learning tool. In summary, our contributions include: a schema classifier trained to predict the relevant schema type and subcategory given a math word problem; structured prompt generation based on the predicted schema/subcategory that uses RAG to include schema-relevant content; a new evaluation metric (the step-by-step reasoning score) to evaluate the quality of the LLM's reasoning steps; and an LLM-as-a-judge evaluation [36].

## 2 Approach

As seen in Figure 1, our approach is divided into four main parts: 1) Schema Classifier, 2) Prompt Creation, 3) Context Retrieval, and 4) Answer and Response Generation. The training and dataset details are described in Appendix C and D, respectively.

A schema is a structured framework that represents a generalized method for solving a specific type of problem [25]. In the context of MWPs, schemas help categorize problems based on their underlying structure, making it easier to determine which mathematical operations to use [9]. For example, MWPs can often be grouped into two major schemas: Additive and Multiplicative [20].

Each schema can be further divided into sub-categories. For instance, the Additive schema can include Additive Change (where a value is increased or decreased), Additive Difference (problems that focus on the difference between two values), and Additive Total (where two or more quantities are combined to get a total) [31]. Similarly, the Multiplicative schema can include Multiplicative Comparison (where one quantity is compared to another using multiplication), Multiplicative Equal Groups (where the total is divided into equal parts), and Multiplicative Ratios/Proportions (problems that involve finding ratios or proportional relationships) [30].

These schemas provide a structured framework for problem-solving, helping both the language model and learners identify the type of problem and apply the appropriate operations.

**Building a Schema Classifier:** We develop a schema classifier that performs supervised learning to predict the relevant schema ($S_i$) and sub-category ($S_{ci}$) for a given problem ($P$). This classifier is built using a DistilBERT model, which has been fine-tuned on a custom dataset of schema-based instruction problems.

Each problem in the dataset is labelled with its associated schema and sub-category, helping the classifier learn the relationships between different types of word problems and their corresponding schemas. Specifically, the input problem is tokenized and processed by the DistilBERT model, which then outputs the most suitable schema ($S_i$) and sub-category ($S_{ci}$).

The schema classifier is essential because it ensures that the correct problem-solving framework is applied to each problem. This step forms the foundation for schema-driven problem solving, guiding the language model to select the appropriate reasoning method.

**Prompt Creation:** Once the schema classifier has predicted the relevant schema ($S_i$) and sub-category ($S_{ci}$), the next stage is prompt creation. This involves generating a structured prompt that instructs the system on how to apply the identified schema to solve the problem. The generated prompt is formulated as follows:

> **"Using the {schema} schema and {sub_category} sub-category, solve the following problem: {question}"**

This prompt guides the language model by ensuring that the problem-solving approach adheres to the appropriate schema.

**Context Retrieval**: After the schema-specific prompt is created, relevant context is retrieved to support the problem-solving process. This retrieval is done using a Retrieval-Augmented Generation (RAG) framework [16], which combines document retrieval with prompt-based generation.

The generated prompt is embedded, and a cosine similarity search [21] is performed within a vector store to identify the most relevant documents [34]. The vector store contains documents that serve as knowledge resources for the problem-solving process. These documents include explanations of various problem-solving strategies, examples of similar problems, and definitions of key mathematical concepts. For instance, a document might explain how to apply the Additive Change schema or provide sample problems involving ratios and proportions.

The document store is particularly useful because it supplies the model with additional knowledge that helps ensure the problem is solved in a structured, context-driven manner. The retrieved documents are ranked based on their similarity to the prompt, ensuring that the most relevant information is used to enhance the problem-solving process [11].

**Answer and Response Generation:** In the final stage, the retrieved context, problem, and schema-specific prompt are passed to the Llama 3.1 LLM [29] for generating the answer. The input is structured by combining the context, schema, and problem, allowing the model to produce a step-by-step solution that incorporates schema-driven reasoning. This ensures that each part of the problem-solving process is addressed in a structured manner, guiding the learner through the solution transparently. The response incorporates relevant contextual information, ensuring that the generated solution is both accurate and aligned with the instructional methodology. This schema-informed and context-enhanced process improves the transparency and effectiveness of the solution.

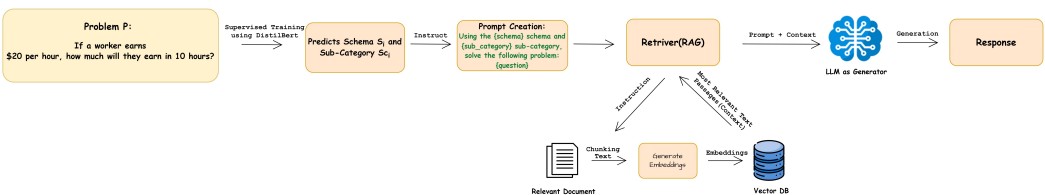

Figure 1: Illustration of SBI-RAG Architecture

# 3 Evaluation

For evaluating the utility of our approach, we focus on the step-by-step reasoning provided by the generated responses, rather than solely on accuracy. Our goal is to ensure that the reasoning process is clear, logical and follows schema-driven methodologies, which helps improve understanding in solving MWPs [25]. To address this, we introduce a new metric, the reasoning score, to measure the quality of the reasoning in the generated solutions. We also evaluate the performance of our schema classifier and analyze both the training and validation losses, ensuring that it generalizes well to unseen data. We also make use of the LLM-as-a-Judge approach [36] to get feedback and evaluate our response from LLMs like GPT-4 and GPT-3.5 Turbo. This approach is a scalable and explainable method for approximating human preferences [14], which are otherwise costly to obtain.

All experiments were run using Google's Colab environment with an NVIDIA L4 GPU. More details on the evaluation and metrics used are given in Appendices D, E, F, and G.

**Schema Classifier Results:** The schema classifier was trained to identify two schema categories and three sub-categories: Additive Change, Additive Difference, Additive Total, Multiplicative Comparison, Multiplicative Equal Groups, and Multiplicative Ratios/Proportions. As seen in Figure 2 and Figure 3, it achieved high precision, recall, and F1 scores, with an overall accuracy of 97%. The training and validation losses show consistent convergence, indicating effective learning without overfitting, ensuring reliable schema predictions across various problem types.

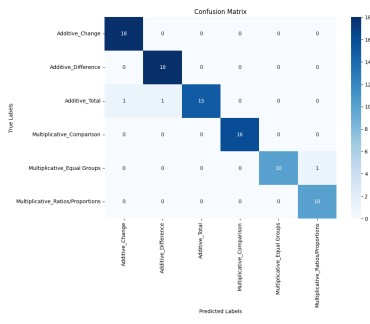

Figure 2: Confusion matrix for the schema classifier

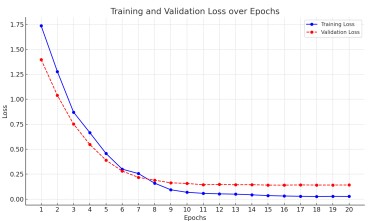

Figure 3: Training and validation losses for the schema classifier

**Reasoning Evaluation** We evaluated the reasoning quality by comparing responses generated using our Schema-Based RAG approach against responses from LLMs like GPT-4 and GPT-3.5 Turbo. The responses generated by our system, which incorporates schema-based reasoning, achieved higher scores in reasoning quality. Specifically, the best reasoning scores for SBI-RAG, GPT-4, and GPT-3.5 Turbo were 0.588, 0.491, and 0.290, respectively. Paired sample t-tests showed that the differences between the SBI-RAG and the GPT models were significantly different at the 0.05 level (see Appendix E). These results suggest that schema-based reasoning can enhance the overall quality of reasoning, particularly in educational contexts, when compared to responses generated by LLMs alone.

**LLM-as-a-Judge Results:** We implemented the LLM-as-a-Judge approach [36] to evaluate the quality of reasoning in the responses generated by both our Schema-Based RAG system and the baseline LLMs. This method allows for an objective, scalable evaluation by approximating human judgment through the use of LLMs. Our LLM-as-a-Judge process involves scoring responses based on clarity, logical progression, and completeness. Results showed that the Schema-Based RAG approach consistently outperformed GPT-4 and GPT-3.5 Turbo in terms of reasoning quality.For more details refer to Appendix G.

## 4  Conclusion

Despite the promising results of our Schema-Based Instruction Retrieval-Augmented Generation (SBI-RAG) framework for improving math word problem reasoning, some limitations exist. This study relies on the LLM-as-a-Judge method, lacking direct human evaluation from educators or students, which would provide more informative feedback. The success of the RAG framework hinges on the relevance and quality of retrieved documents, which may vary and impact the generated solutions. The evaluation focuses on arithmetic word problems (GSM8K). More complex problem datasets are needed to assess the framework's generalizability. Finally, extending the framework to different subjects or educational levels may present challenges, requiring further adaptation. These limitations highlight areas for future research, particularly in improving schema coverage, expanding dataset diversity, and incorporating human evaluations.

In conclusion, we proposed a Schema-Based Retrieval-Augmented Generation framework that enhances reasoning and understanding in solving math word problems. Our approach, combining schema-based instruction with large language models, outperformed existing LLM responses in

quality and step-by-step reasoning. This framework provides a strong foundation for improving problem-solving in education, with future work focused on refining the system with user feedback. Additionally, this work could have applications in enhancing the reasoning capabilities of LLMs themselves.

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

# A    Appendix / supplemental material

# B    Related Work

## B.1    Intelligent Tutor System

Intelligent Tutoring Systems (ITS) [10] have been widely adopted to provide personalized learning experiences for students. ITS systems aim to emulate human tutors by guiding learners through problem-solving activities and providing timely feedback. One such system, **HINTS** [35], focuses on helping students navigate through mathematical problem-solving by offering hints and scaffolding to improve their understanding and success rates. The system is designed to foster incremental learning through step-by-step hints tailored to the learners' needs, improving problem-solving skills over time.

## B.2    MWPTutor

**MWPTutor** [5], a system developed for solving Math Word Problems (MWPs), integrates schema-based instruction into its design. It provides structured guidance to students by breaking down word problems into solvable chunks using predefined schemas for addition, subtraction, multiplication, and division. The tutor system helps students plan their solutions and execute them using step-by-step guidance while providing immediate feedback. **MWPTutor** incorporates interactive elements like highlighting important information in word problems and constructing solution trees to visualize the solution path. These features improve students' understanding of both the problem and the solution process.

## B.3    MWP-BERT

Recent advancements in **Math Word Problem (MWP)** [17] solving have leveraged large pre-trained language models such as BERT. **MWP-BERT**, a numeracy-augmented model, addresses a significant challenge in MWP solving—efficient numerical reasoning. Traditional models struggle with representing numbers accurately, often substituting real numbers with symbolic placeholders, which overlooks crucial numerical properties. MWP-BERT introduces a novel pre-training schema that incorporates numerical reasoning into the language model. It enhances the model's ability to generalize over arithmetic and algebraic problems by embedding numeracy information such as magnitude and number types into contextualized word representations. This approach has outperformed many conventional MWP solvers, especially in arithmetic MWP datasets like Math23k [32] and MathQA [1], by accurately capturing the logic of number manipulation within word problems.

### B.4 HINTS

The **HINTS** [35] system emphasizes providing incremental and context-sensitive support to students working on math problems. This tutor-like system offers "hints" that gradually lead students toward the correct solution without directly giving them the answer. This method allows students to develop their problem-solving strategies while avoiding the frustration of being stuck. The system also records students' problem-solving paths to provide personalized feedback, allowing for the analysis of specific challenges faced during different stages of the solution process.

### B.5 MathCal

**MathCAL** [4] is another example of a computer-assisted learning system designed to support mathematical problem-solving. It operates by dividing the problem-solving process into four distinct stages: understanding the problem, making a plan, executing the plan, and reviewing the solution. Each stage provides specific assistance tailored to the learner's needs, with tools such as schema representations and solution trees helping students visualize and articulate their solution process . The empirical evaluation of MathCAL demonstrated its effectiveness in improving problem-solving performance, particularly among students with lower baseline abilities. By breaking down complex problems into manageable steps, MathCAL reduces cognitive load and promotes deeper understanding of mathematical concepts.

## C  Datasets

### C.1  Schema Based Instruction Dataset

The Schema-Based Instruction (SBI) Dataset consists of a total of 360 math word problems (MWPs), categorized based on their underlying schemas. These problems are distributed equally across six distinct categories, with approximately 60 problems in each sub-category (as seen in figure 4) . The categories include Additive Change, Additive Difference, Additive Total, Multiplicative Comparison, Multiplicative Equal Groups, and Multiplicative Ratios/Proportions.

Each problem is labeled with a schema (Additive or Multiplicative) and its corresponding sub-category, allowing the system to learn and predict the appropriate schema for a given problem. This balanced distribution ensures that the model receives equal representation from each schema type, preventing overfitting to any specific category and promoting generalization across diverse problem types.

This dataset is used to learn the relationship between a given MWP and its corresponding schema. By using this dataset, a schema classifier is trained to accurately predict the appropriate schema for each problem. This classifier plays a crucial role in facilitating schema-based retrieval and guiding the generation of step-by-step solutions, ensuring clarity and structure in the problem-solving process.

### C.2  GSM8K Dataset

GSM8K is a dataset of 8.5K high quality linguistically diverse grade school math word problems created by human problem writers. The dataset is segmented into 7.5K training problems and 1K test problems. These problems take between 2 and 8 steps to solve, and solutions primarily involve performing a sequence of elementary calculations using basic arithmetic operations to reach the final answer. A bright middle school student should be able to solve every problem. It can be used for multi-step mathematical reasoning [6].

## D  Training Details and Implementation

The code for this implementation can be found on GitHub: `https://github.com/pdx97/SBI-RAG_Neurips2024`.

### D.1  Dataset and Preprocessing

The dataset used for training consists of schema-based instruction (SBI) problems, where each problem is labeled with a schema and a sub-category. These labels are combined into a single label

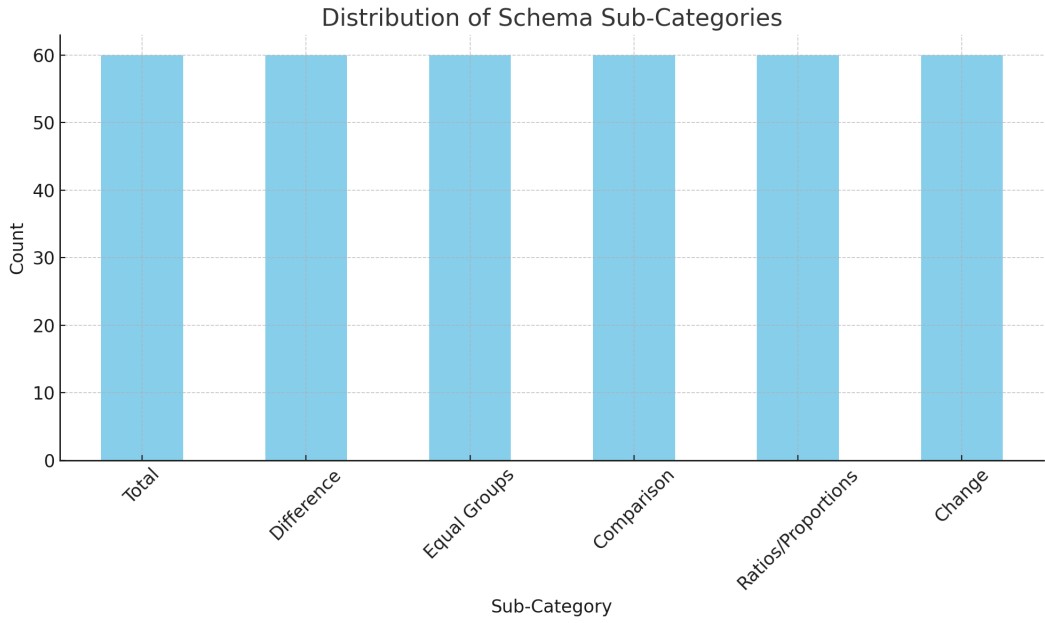

Figure 4: Overview of SBI Dataset

**Problem:** Beth bakes 4, 2 dozen batches of cookies in a week. If these cookies are shared amongst 16 people equally, how many cookies does each person consume?
**Solution:** Beth bakes 4 2 dozen batches of cookies for a total of 4*2 = <<4*2=8>>8 dozen cookies
There are 12 cookies in a dozen and she makes 8 dozen cookies for a total of 12*8 = <<12*8=96>>96 cookies
She splits the 96 cookies equally amongst 16 people so they each eat 96/16 = <<96/16=6>>6 cookies
**Final Answer:** 6

**Problem:** Mrs. Lim milks her cows twice a day. Yesterday morning, she got 68 gallons of milk and in the evening, she got 82 gallons. This morning, she got 18 gallons fewer than she had yesterday morning. After selling some gallons of milk in the afternoon, Mrs. Lim has only 24 gallons left. How much was her revenue for the milk if each gallon costs $3.50?
Mrs. Lim got 68 gallons - 18 gallons = <<68-18=50>>50 gallons this morning.
So she was able to get a total of 68 gallons + 82 gallons + 50 gallons = <<68+82+50=200>>200 gallons.
She was able to sell 200 gallons - 24 gallons = <<200-24=176>>176 gallons.
Thus, her total revenue for the milk is $3.50/gallon x 176 gallons = $<<3.50*176=616>>616.
**Final Answer:** 616

**Problem:** Tina buys 3 12-packs of soda for a party. Including Tina, 6 people are at the party. Half of the people at the party have 3 sodas each, 2 of the people have 4, and 1 person has 5. How many sodas are left over when the party is over?
**Solution:** Tina buys 3 12-packs of soda, for 3*12= <<3*12=36>>36 sodas
6 people attend the party, so half of them is 6/2= <<6/2=3>>3 people
Each of those people drinks 3 sodas, so they drink 3*3=<<3*3=9>>9 sodas
Two people drink 4 sodas, which means they drink 2*4=<<4*2=8>>8 sodas
With one person drinking 5, that brings the total drank to 5+9+8+3= <<5+9+8+3=25>>25 sodas
As Tina started off with 36 sodas, that means there are 36-25=<<36-25=11>>11 sodas left
**Final Answer:** 11

Figure 5: GSM8K dataset example problems

for multi-class classification. The dataset is split into training and testing sets, with 75% of the data used for training and 25% for testing.

The dataset is tokenized using the `distilbert-base-uncased` tokenizer from Hugging Face, and the text is converted into input tensors consisting of `input_ids` and `attention_masks`. Label encoding is applied to the combined schema and sub-category labels using `LabelEncoder` from `sklearn`. The resulting dataset is then formatted for PyTorch, with columns for input IDs, attention masks, and labels.

## D.2    Model Architecture for Schema Classifier

We used the DistilBERT model for schema classification, loaded from Hugging Face's `transformers` library. The model is pre-trained and fine-tuned on our custom SBI dataset. The number of output labels is set to the number of unique schema and sub-category combinations in the dataset.

## D.3    Training Process

The training was conducted using the `Trainer` API from Hugging Face [13] with the configuration shown in Table 1.

| Hyperparameter | Value |
|---|---|
| Learning rate | $2 \times 10^{-5}$ |
| Batch size | 16 |
| Number of epochs | 20 |
| Optimizer | AdamW with weight decay of 0.01 |
| Evaluation strategy | Model evaluation at the end of each epoch |
| Logging | Evaluation results logged every 10 steps |

Table 1: Training Hyperparameters for Schema-Based Classifier

## D.4    Evaluation and Results

As seen in Figures 4 and 5, we evaluated the schema classifier using accuracy, precision, recall, F1-scores, and a confusion matrix. The classifier achieved an overall accuracy of 97%. The training and validation losses show consistent convergence, indicating effective learning without overfitting, ensuring reliable schema predictions across various problem types.

## D.5    Context Retrieval Implementation

Once the schema and sub-category are predicted, the next step involves retrieving relevant context for solving the problem. The document source is loaded from a URL (`https://iris.peabody. vanderbilt.edu/module/math/cresource/q2/p06/`) [27] using the `WebBaseLoader`. The loaded text is split into chunks of 1000 characters with an overlap of 200 characters to ensure completeness of context during retrieval.

## D.6    Vector Store and Embeddings

We use Ollama embeddings to create document embeddings for context retrieval. The embeddings are stored in a Chroma vector store. During the retrieval process, the problem and schema-specific prompt are embedded, and a similarity search is performed to retrieve the most relevant documents. Re-ranking of documents is performed based on cosine similarity between the embedded question and the retrieved documents.

## D.7    Response Generation with Ollama Llama 3.1

For generating a solution to the problem, we pass the schema, sub-category, and retrieved context to the Llama 3.1 model. The prompt is constructed in a structured format, and the model generates a detailed solution based on the provided context.

## D.8 Advanced Re-ranking for Document Retrieval

An advanced re-ranking mechanism is implemented using cosine similarity between question embeddings and document embeddings. This ensures that the most contextually relevant documents are used for generating the final answer.

## E Statistical Significance

To test the statistical significance, a paired sample t-test, also known as a dependent sample t-test, was conducted to compare the reasoning performance of SBI-RAG with two language models, GPT 3.5 Turbo and GPT 4.0. The paired sample t-test is important because it compares the means of two sets of measurements taken from the same subjects or related units. In this case, the same set of problems was evaluated using both SBI-RAG and the GPT models, meaning the samples are dependent. By using this approach, we can account for the relationship between the scores, reducing variability and making the comparison more accurate.

The results showed that SBI-RAG reasoning scores were statistically higher than both GPT 3.5 Turbo and GPT 4.0. For the comparison with GPT 3.5 Turbo, the t-test gave a t-statistic of 5.87 and a p-value of 0.00012, which is much lower than the 0.05 threshold. Similarly, for the comparison with GPT 4.0, the t-test gave a t-statistic of 3.69 and a p-value of 0.00248, also well below 0.05.

These results confirm that SBI-RAG outperforms both GPT 3.5 Turbo and GPT 4.0 in reasoning tasks. With p-values much lower than 0.05, we can confidently reject the null hypothesis, which assumed no difference in performance, and conclude that SBI-RAG consistently achieves higher reasoning scores.

## F Reasoning Score Metric and Implementation

**Reasoning Score Metric**

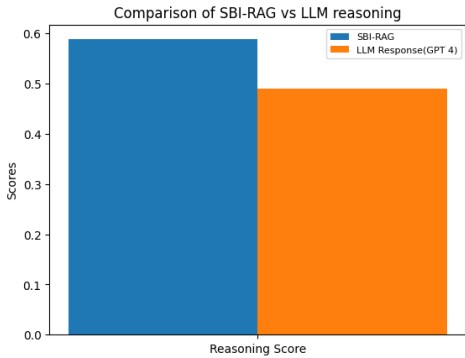 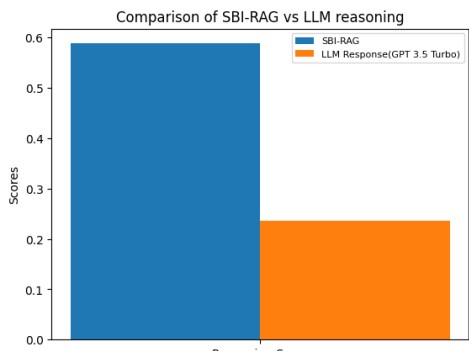

Figure 6: Reasoning Score SBI-RAG vs GPT-4

Figure 7: Reasoning Score SBI-RAG vs GPT 3.5 Turbo

The reasoning score is calculated by checking both the presence of key steps and the logical flow between them. We first define a set of key steps and concepts relevant to solving the problem, such as operations ("+", "*", "-"), schema-related terms ("Additive", "Multiplicative"), and problem-specific concepts ("ratios", "proportions"). We then count how many of these key steps appear in the generated response. In addition to counting the presence of steps, we calculate a delta score, which checks the logical flow between steps.

For example, consider the problem:

> *Each bird eats 12 beetles per day, each snake eats 3 birds per day, and each jaguar eats 5 snakes per day. If there are 6 jaguars in a forest, how many beetles are eaten each day?*

In this problem, the key steps include calculating how many snakes are eaten by the jaguars, how many birds are eaten by the snakes, and how many beetles are eaten by the birds. The delta score evaluates whether the transitions between these entities are correctly captured in the reasoning, for example:

- The transition from "jaguars" to "snakes" (i.e., each jaguar eats 5 snakes per day).
- The transition from "snakes" to "birds" (i.e., each snake eats 3 birds per day).
- The transition from "birds" to "beetles" (i.e., each bird eats 12 beetles per day).

The final reasoning score is computed by combining the step-matching score with the delta score to account for both completeness and logical progression. The score is further adjusted by a clarity factor, which depends on the length and clarity of the explanation. A higher clarity factor indicates a more detailed and structured response. For instance, in this problem, a well-reasoned response would clearly explain how the total number of snakes eaten by jaguars leads to the calculation of the total number of beetles eaten by birds.

# G  LLM-as-a-Judge Results and Task Definition

LLM-as-a-Judge is a reference-free evaluation method that leverages large language models (LLMs) to score and evaluate the quality of generated responses. This approach is particularly useful when human evaluation is costly or impractical to scale. By directly prompting an LLM to assess the reasoning, clarity, and structure of an answer, we can measure how well the response aligns with human preferences. Our study follows this methodology to evaluate the quality of schema-based reasoning responses in math word problems (MWPs).

**Task Design**: We designed the evaluation prompt based on guidelines from Hugging Face's LLM-as-a-Judge model (as seen in Figure 8). Our customized prompt asked the LLM to act as a judge and evaluate responses from an educational perspective. The system was tasked to rate each response on a scale of 0 to 10, where 0 meant the response was not helpful at all, and 10 meant the response was complete and thoroughly addressed the question. The rating also considered the clarity and educational effectiveness of the responses.

The task was defined using the following prompt template:

```
You will be given a user_question and system_answer couple.
Your task is to provide a 'total rating' scoring how well the system_answer
answers the user concerns expressed in the user_question.
Give your answer as a float on a scale of 0 to 10, where 0 means that the system_answer
is not helpful at all, and 10 means that the answer completely and helpfully addresses
the question.

Provide your feedback as follows:

Feedback:::
Total rating: (your rating, as a float between 0 and 10)

Now here are the question and answer.

Question: {question}
Answer: {answer}
Feedback:::
Total rating:
```

**Evaluation Results**: As shown in Figure 11 and Figure 12, two responses were evaluated based on a math word problem: *"James spends 40 years teaching. His partner has been teaching for 10 years less. How long is their combined experience?"*.

**Response 1**  provided a solution that followed a schema-based approach, utilizing the Additive schema and the Total sub-category. It offered a clear and detailed step-by-step explanation, guiding

[Task]
You will be given a user_question and Responses for that question. You have to act as a Judge and evaluate those responses from an educational point of view.

Your task is to provide a 'total rating' scoring to each response and how well the system_answer answers the user concerns expressed in the user_question and whether they follow step by step reasoning with clarity
Give your answers as a float on a scale of 0 to 10, where 0 means that the responses is not helpful at all, and 10 means that the answer completely and helpfully addresses the question from an educational point of view.

Feedback:::
Total rating: (your rating, as a float between 0 and 10)

Now here are the question and responses.

Figure 8: LLM-as-a-Judge Task Instructions for Evaluating Responses.

the reader through each part of the process. The response emphasized the use of schema-driven reasoning to help break down the problem and apply the correct operations, making it highly suitable for educational purposes. The structured reasoning and clarity of explanation were acknowledged by the judge, who rated this response highly, giving it a score of **9.5/10** for its thoroughness and educational value.

**Response 2** also arrived at the correct solution but lacked the same depth of explanation. It skipped several intermediate steps and did not provide a schema-based breakdown of the problem, making it less effective from an educational standpoint. While it was concise and accurate, it did not fully guide the learner through the reasoning process, which reduced its value for students needing additional support. Consequently, the judge assigned this response a score of **8.5/10**, noting that while it was correct, it could benefit from more detailed reasoning and a clearer breakdown of steps.

Additionally, For a given Question (As seen in Figure9), our evaluation using the LLM-as-a-Judge approach assessed responses based on three key sub-metrics: *Clarity*, *Logical Progression*, and *Completeness* as shown in Figure 10. The Total rating was then calculated based on these sub-metrics, which are defined as follows:

**[Question]**
A parking lot has 8 rows with 10 cars in each row. How many cars are there in the parking lot?
**[Response 1]**
Schema used: Multiplicative
Sub-category used: Equal Groups
Answer: Here's how to solve the problem using the Multiplicative schema and Equal Groups sub-category:

Problem: A parking lot has 8 rows with 10 cars in each row. How many cars are there in the parking lot?

Solution Steps:

1. Identify the number of rows (groups) and the number of cars in each row.
* Number of rows: 8
* Number of cars per row: 10
2. Set up a multiplication equation using the Equal Groups schema.
* Equation: 8 x 10 = ?
3. Multiply the number of rows by the number of cars per row.
* Calculation: 8 x 10 = 80

Answer:

There are 80 cars in the parking lot.

**[Response 2 ]**
To find the total number of cars in the parking lot, we need to multiply the number of rows by the number of cars in each row.

8 rows * 10 cars = 80 cars

Therefore, there are 80 cars in the parking lot.

Figure 9: Sample question with Response 1 from SBI-RAG and Response 2 from GPT-4

- **Clarity:** Assesses how clearly the response conveys the solution. A high clarity score indicates ease of understanding, appropriate language use, and avoidance of unnecessary jargon or complexity that could confuse the reader.

- **Logical Progression:** Evaluates the logical flow of the response. A high score here indicates that each step follows naturally from the previous one, forming a coherent sequence that effectively guides the reader through the problem-solving process.

- **Completeness:** Measures whether the response fully addresses all aspects of the question. A complete response includes all necessary steps, explanations, and justifications required to reach the solution.

| Response | Clarity (0-10) | Logical Progression (0-10) | Completeness (0-10) | Total Rating (0-10) |
|---|---|---|---|---|
| Response 1 | 9.0 | 9.0 | 9.0 | 9.0 |
| Response 2 | 8.0 | 7.5 | 7.0 | 7.5 |

Table 2: Evaluation Scores for Response 1 and Response 2

The judge was able to provide feedback explaining why each response received its respective score(as seen in Fig 13 . This structured feedback highlighted the strengths of schema-based reasoning in fostering better understanding and logical problem-solving, especially when compared to answers that merely focused on arriving at the correct solution without explaining intermediate steps.

**Response 1:**

- **Clarity**: Scored 9.0 for its detailed and easy-to-understand explanation using schema terms. The response guides the reader through each calculation step clearly.
- **Logical Progression**: Scored 9.0 as it follows a logical, step-by-step approach that effectively breaks down the problem.
- **Completeness**: Scored 9.0 for covering all steps in the solution, including schema-based reasoning, which adds educational depth.

**Total Rating**: 9.0
**Feedback**: This response is highly educational and detailed, providing clarity, a structured approach, and schema-based insight.

**Response 2:**

- **Clarity**: Scored 8.0 for its straightforward explanation, although it is less detailed than Response 1.
- **Logical Progression**: Scored 7.5 as it presents a logical calculation sequence but is brief and lacks structured reasoning.
- **Completeness**: Scored 7.0, as it provides the correct answer but lacks the schema-based explanation, making it less educational.

**Total Rating**: 7.5
**Feedback**: This response is concise and accurate but could benefit from additional explanation and schema-based reasoning to enhance its educational value.

Figure 10: Overall Scores by LLM-as-a-Judge

**[Question]**
James spends 40 years teaching. His partner has been teaching for 10 years less. How long is their combined experience?

**[Response 1]**
Schema used: Additive
Sub-category used: Total
Answer: Here's the solution using the Additive schema and Total sub-category:

**Problem:** James spends 40 years teaching. His partner has been teaching for 10 years less. How long is their combined experience?

**Step-by-Step Solution:**

1. Identify the total number of years that James has been teaching: 40 years
2. Identify the difference in years between James and his partner: 10 years (since his partner has been teaching for 10 years less)
3. Add the difference to one of the numbers to find the other person's experience: His partner has been teaching for 40 - 10 = 30 years
4. Calculate their combined experience by adding both numbers: James' experience + partner's experience = 40 + 30 = 70 years

**[Response 2]**
If James has been teaching for 40 years, and his partner has been teaching for 10 years less, then his partner has been teaching for 40 - 10 = 30 years.

Therefore, their combined experience is 40 + 30 = 70 years.

Figure 11: Response 1 is using SBI-RAG and Response 2 is using GPT 3.5 turbo.

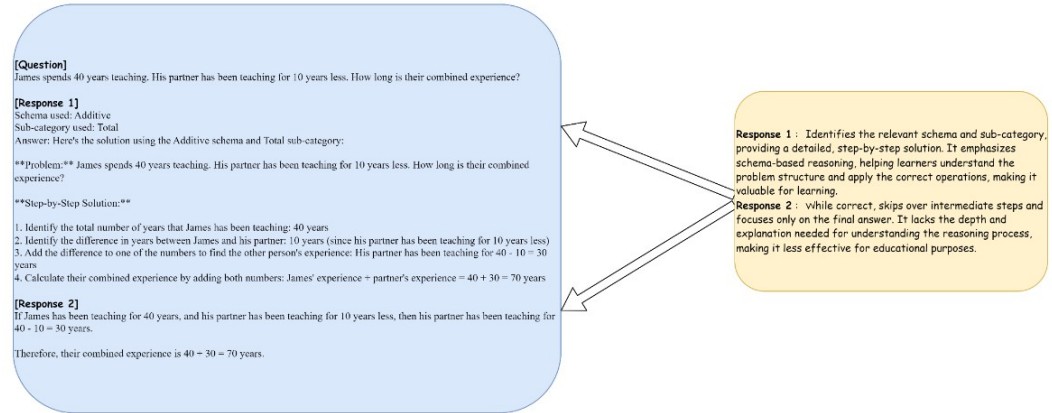

Figure 12: Response 1 and Response 2 evaluation

Our results demonstrate the effectiveness of using LLM-as-a-Judge for assessing educational content. The schema-driven responses generated by our system scored higher in terms of educational effectiveness and reasoning quality, emphasizing the potential of schema-based approaches in improving learning outcomes in math word problems.

## Feedback:

**Response 1**
- **Total rating**: 9.5
- **Evaluation**: This response provides a clear, well-structured solution using the Additive schema and Total sub-category, which is helpful from an educational point of view. The explanation is thorough, breaking down each step logically and guiding the student through the process of finding the answer. The use of schema adds educational value and helps reinforce mathematical reasoning. The only minor improvement could be simplifying the phrasing slightly to be even more concise, but overall, it is an excellent response.

**Response 2**
- **Total rating**: 8.5
- **Evaluation**: This response is accurate, concise, and easy to understand. It clearly explains the calculation, but it lacks the depth of schema-based explanation and step-by-step breakdown that would enhance its educational value. While it provides a correct answer, it might not help students who need more guidance in understanding the reasoning behind the solution. A bit more explanation would improve its educational effectiveness.

Figure 13: Feedback of Response 1 and Response 2

By utilizing this method, we can approximate human preferences and make informed decisions about how schema-based approaches can enhance student learning experiences in classrooms. Future work could extend this by incorporating human feedback and further refining the evaluation process.

