# OpenReview forum: "SBI-RAG: Enhancing Math Word Problem Solving for Students through Schema-Based Instruction and Retrieval-Augmented Generation."
_NeurIPS.cc/2024/Workshop/MATH-AI — MATH-AI 24_

### Official Review · Reviewer_vRBp · 2024-10-03
**A Review of SBI-RAG: Enhancing Math Word Problem Solving through Schema-Based Instruction and Retrieval-Augmented Generation**

**Rating:** 7
**Confidence:** 4

**Review:**

The paper presents a novel framework, Schema-Based Instruction Retrieval-Augmented Generation (SBI-RAG), aimed at improving students' ability to solve math word problems (MWPs) by leveraging schema-based instruction and large language models (LLMs). The authors propose a structured approach that combines schema classification, context retrieval, and LLM-based solution generation to enhance reasoning clarity and problem-solving accuracy. The framework is evaluated using the GSM8K dataset, and a new "reasoning score" metric is introduced to assess the quality of solutions.

#### Quality and Clarity:
The paper is well-structured and clearly articulates the motivation behind the proposed framework. The introduction provides a comprehensive overview of the challenges students face with MWPs and the potential of schema-based instruction to address these issues. The methodology is detailed, with each component of the SBI-RAG framework explained thoroughly. The evaluation section presents quantitative results that support the claims of improved reasoning quality.

#### Originality:
The integration of schema-based instruction with retrieval-augmented generation is a novel approach in the context of educational technology. While schema-based instruction is a well-established method, its combination with LLMs and retrieval-augmented generation represents an innovative application that could have significant implications for educational practices.

#### Significance:
The proposed framework has the potential to significantly impact educational practices by providing a more structured and effective method for teaching MWPs. The use of LLMs to enhance schema-based instruction could lead to improved problem-solving skills and deeper understanding among students. The introduction of a reasoning score metric also contributes to the field by providing a new way to evaluate the quality of reasoning in generated solutions.

#### Pros:
- **Innovative Approach**: The combination of schema-based instruction with LLMs and retrieval-augmented generation is a novel and promising approach.
- **Comprehensive Evaluation**: The use of a reasoning score metric and LLM-as-a-Judge evaluation provides a robust assessment of the framework's effectiveness.
- **Clear Methodology**: The paper provides a detailed explanation of the framework's components, making it easy to understand and replicate.

#### Cons:
- **Lack of Human Evaluation**: The reliance on LLM-as-a-Judge for evaluation, without direct human feedback from educators or students, is a limitation. Human evaluations could provide more nuanced insights into the framework's effectiveness.
- **Dataset Limitations**: The evaluation is limited to the GSM8K dataset, which focuses on arithmetic word problems. The framework's generalizability to more complex problems or different educational levels is not assessed.
- **Dependence on Document Quality**: The success of the RAG framework depends on the quality and relevance of retrieved documents, which could vary and impact the generated solutions.

---

### Official Review · Reviewer_37oB · 2024-10-06

**Rating:** 6
**Confidence:** 2

**Review:**

### Summary:

The paper proposes an approach for solving math word problems (MWPs) by incorporating Schema-Based Instruction (SBI) with an LLM for enhanced reasoning. The authors have introduced a Schema-Based Instruction Retrieval-Augmented Generation (SBI-RAG) framework, which combines the strengths of SBI and the capabilities of an LLM

### Strengths:

1. SBI is employed to categorize word problems based on their underlying structure (e.g., additive or multiplicative schemas), which helps students and the LLM focus on the appropriate problem-solving method.
2. The authors enhance the prompt generation by retrieving relevant documents or schemas that help guide the LLM's reasoning process. To the best of my understanding, this RAG approach ensures that the model is contextually aware of both the problem and possible solutions.
3. The framework promotes structured reasoning, which is evaluated by the “reasoning score” metric introduced. This metric assesses the quality of stepwise reasoning, rather than just the correctness of the final answer.

### Strengths:

- The use of SBI helps break down problems in a structured manner, making it easier for LLMs to identify and apply the correct operations.
- The evaluation method, LLM-as-a-Judge, is used to assess the quality of the responses, making the evaluation scalable and aligned with human preferences.

### Weaknesses:

- The quality of retrieved documents heavily influences the generated solutions, and any limitations in the document set can affect the reasoning quality.
- The study primarily focuses on arithmetic word problems. To assess the generalizability of the framework, more complex problem datasets need to be included.
- Although LLM-as-a-Judge is a scalable evaluation method for large datasets, I am always a bit skeptical of approaches that use LLMs to verify the outputs of LLMs. What are the guarantees of the output of the
- The full results were not in the main paper. I had to refer the appendix to get a better sense of the results.

### Questions:

1. How would the SBI-RAG framework handle more advanced MWPs involving algebra, calculus, or geometry, which might require more complex schemas?
2. Can you elaborate on how document retrieval errors (e.g., irrelevant or incomplete information) are mitigated? How sensitive is the system to these types of errors?
3. How do you plan to incorporate feedback from students and teachers to improve the system in future iterations?
4. Would human evaluations from students and educators align with the LLM-as-a-Judge results? Have you considered pilot testing the system in an educational setting?

---

### Official Review · Reviewer_Jwhu · 2024-10-08
**Novel framework for math word problem teaching shows potential, but flaws in evaluation warrant rejection; further work needed, especially on evaluation.**

**Rating:** 4
**Confidence:** 4

**Review:**

In general I think the authors proposed an interesting framework to enhance math word problem teaching. While it might have promising potentials to help students, I believe the paper is not yet ready to be accepted. I encourage authors to work on this system further, especially on the evaluation part.


## Pros

1. The authors proposed a novel approach to address the MWP solving problem. The framework was complex yet clearly and concisely presented.

2. The paper was mostly easy to read and comprehend. The data and model used are clearly described.

3. The authors recognized the importance of incorporating user feedback and proposed to work on this as future extensions.

4. The authors proposed a new evaluation metric (step-by-step reasoning score) to evaluate the "reasoning" perspective of LLM responses, which resonates with partial credits granted by teachers to students in math problem solving.


## Cons

### Approach

There are some interesting pieces of the system that are not discussed, some of which seem important to the success of the sytem.

1. As part of RAG process, authors generate embedding out of prompt "Using the {schema} schema and {sub_category} sub-category, solve the following problem: {question}" and search the vector store. The authors didn't explain how and why they choose this natural language approach over, say, embedding of "{schema}|{sub_category}|{question}". Further, given the fact that there are only 2 schemas and 3 subcategories, why they decided not to create 6 vector databases (2 schemas × 3 subcategories) and use exact match for schema and sub_category rather than embeddings.

2. Authors didn't mention the prompt template they use in their approach and in baseline. In figure 9 they provided an example pair of responses given by their approach and by baseline, which helped with illustration a lot, but the prompt template wasn't provided. This seems very important to me, especially because the baselines used pure prompting approaches, and different prompts could make a big difference in terms of "reasoning clarity".


### Evaluation Methodology

Overall, the authors didn't convince me their approach is better than pure prompting approaches.

1. The abstract claimed the proposed approach enhanced "problem-solving accuracy" but there's no evaluation on accuracy presented in the paper.

2. Baseline only used GPT-4 and GPT-3.5-turbo, while stronger models like GPT4-turbo released in November, 2023 and GPT4o released in May, 2024, were not used.

3. L417 mentioned the reasoning score was computed by counting number of key steps appeared in the generated response. I noticed the keys included "schema-related terms". From Figure 9 we can see the proposed approach **seemingly** included `Schema-used:` in the prompt template, which suggests the reasoning score metric potentially unfairly favored the proposed approach over baselines.

4. L154 mentioned `More details about the implementation and results can be found in Appendix G`. Appendix G, however, only talked about implementation and results for a sample task. The overall scores were not provided.

5. The LLM-as-a-Judge method uses GPT as a judge. One criterion was "whether they follow step by step reasoning with clarity". Since the exact prompt templates for baseline and proposed approach were not provided, we can only guess from the sample responses in Figure 9. It **seems** the proposed approach included `step-by-step solution` in the prompt while the baseline may or may not have included similar prompts. Since the evaluation is based on whether they follow `step by step` reasoning, it seems unfair if the baselines did not use CoT prompting technique.

### Writings

There are a few small flaws in writing that could be improved, but they are in general minor.

1. The prompt template for LLM-as-a-Judge evaluation in L452-458 is different from figure 8.

2. L130 used ambiguous wording: "LLMs **like** GPT-4 and GPT-3.5 Turbo". Using wording like "including" would be more accurate since this piece of text is talking about evaluation, where precision and clarity are crucial.

3. L47 cited paper [22] "Hannah Rashkin, Vitaly Nikolaev, Matthew Lamm, Lora Aroyo, Michael Collins, Dipanjan Das, Slav Petrov, Gaurav Singh Tomar, Iulia Turc, and David Reitter. Measuring attribution in natural language generation models. Computational Linguistics, 49(4):777–840, 2023." for their criticism of CoT prompting. To my best knowledge, this reference does not cover CoT prompting.

---

### Decision · Program_Chairs · 2024-10-09

**Decision:**

Accept

**Comment:**

The approach is interesting, though we encourage the authors to carefully consider the weaknesses pointed out by Reviewer Jwhu to more convincingly highlight the strengths of this methodology.